# Adaptive Identification of Blurred Regions for Accurate Image Deblurring

## Abstract

Image deblurring aims to restore high-quality images from blurred ones. While existing deblurring methods have made significant progress, most overlook the fact that the degradation degree varies across different regions. In this paper, we propose AIBNet, a network that adaptively identifies the blurred regions, enabling differential restoration of these regions. Specifically, we design a spatial feature differential handling block (SFDHBlock), with the core being the spatial domain feature enhancement module (SFEM). Through the feature difference operation, SFEM not only helps the model focus on the key information in the blurred regions but also eliminates the interference of implicit noise. Additionally, based on the fact that the difference between sharp and blurred images primarily lies in the high-frequency components, we propose a high-frequency feature selection block (HFSBlock). The HFSBlock first uses learnable filters to extract high-frequency features and then selectively retains the most important ones. To fully leverage the decoder's potential, we use a pre-trained model as the encoder and incorporate the above modules only in the decoder. Finally, to alleviate the resource burden during training, we introduce a progressive training strategy. Extensive experiments demonstrate that our AIBNet achieves superior performance in image deblurring. Our code is available at https://anonymous.4open.science/r/AIBNet-541B/

## 1 Introduction

Image deblurring aims to remove blur and restore clean images. Due to the ill-posed nature of the problem, traditional methods Karaali & Jung (2017) try to tackle it by introducing priors to constrain the solution space. However, formulating these priors is difficult and often lacks broad applicability, making them unsuitable for real-world scenarios.

With the rapid advancement of deep learning, convolutional neural networks (CNNs) Cui et al. (2024); Mao et al. (2024) have become the preferred approach for image deblurring. They excel at implicitly learning generalized priors by capturing natural image statistics, achieving state-of-the-art performance. However, while convolutional operations are effective at modeling local connections, their limited receptive field and inability to adapt to input content restrict the model's ability to capture long-range dependencies. To overcome these limitations, Transformers Li et al. (2025); Feng et al. (2024) have been incorporated into image deblurring. With their self-attention mechanism and adaptive weights, Transformers capture global dependencies more effectively, outperforming CNN-based methods. More recently, Mamba-based networks Liu et al. (2025b); Guo et al. (2025) have been applied to image deblurring tasks. These networks capture global information with linear complexity, offering greater efficiency than Transformers.

Although the methods mentioned above have achieved excellent performance through modular design, most overlook the fact that **the degradation degrees varies across different regions of the blurred image.** Treating all regions as having the same degree of degradation inevitably leads to the introduction of artificial artifacts in the restored image. As shown in Figure 1, (a) is the blurred image, (b) is the corresponding clear image, and (c) is the residual image between the blurred and clear images. It is evident that the areas marked with red boxes are more heavily degraded, while those marked with green boxes exhibit less degradation. The residual images also highlight that the difference between the clear and blurred image pairs is nearly zero in the less degraded regions. To enable

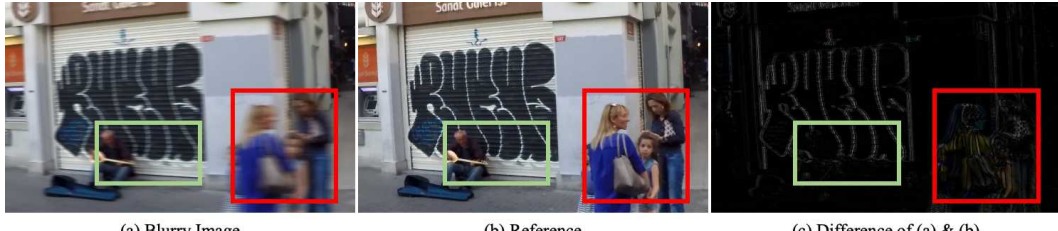

(a) Blurry Image  (b) Reference  (c) Difference of (a) & (b)

Figure 1: Varying degrees of degradation across different regions. (a) is the blurred image, (b) is the corresponding clear image, and (c) is the residual image between the blurred and clear images.

differential handling of varying degrees of degradation, AdaRevD Mao et al. (2024) introduces a classifier to assess the degradation degree of image patches. However, this method categorizes patches into six degradation levels based on the PSNR between the blurred and clear patches, and uses a fixed, relatively large patch size of 384 x 384 for classification. This rigid approach reduces its effectiveness in adaptively managing degradation across patches of different sizes.

Based on the above analysis, we are motivated to find a method that can adaptively handle regions with varying degrees of degradation. To achieve this, we propose AIBNet, a network that adaptively identifies blurred regions in both the spatial and frequency domains. Specifically, we design a spatial feature differential handling block (SFDHBlock), consisting of a spatial feature enhancement module (SFEM) and a simple channel attention (SCA) Chen et al. (2022). Drawing from the theory of differential amplifiers, SFEM uses feature differences to remove features from non-blurred regions and reduce implicit noise caused by intensive calculations, helping the model focus on key information in the blurred regions. Meanwhile, we use the SCA to capture spatial domain features. The features from SFEM and SCA are fused using learnable weights, enhancing the representation of features in the blurred regions.

Additionally, recognizing that the difference between clear/blurred images primarily lies in the high-frequency components, we present a high-frequency feature selection block (HFSBlock). The HFS-Block first uses learnable filters to extract high-frequency features, then selectively retains the most important high-frequency information to emphasize the features of the degraded regions. To fully leverage the potential of the decoder, we use a pre-trained model as the encoder and adopt multiple sub-decoders. Finally, to reduce the resource burden during training, we introduce a progressive training strategy.

The main contributions of this work are:

1. We propose an adaptively identifies blurred regions network (AIBNet) for image deblurring. Extensive experiments demonstrate that the proposed AIDNet achieves promising performance across synthetic and real-world datasets.

2. We design a spatial feature differential handling block (SFDHBlock), with the core being the spatial feature enhancement module (SFEM). SFEM uses feature differences to help the model focus on key information in the blurred regions.

3. We present a high-frequency feature selection block (HFSBlock) that extract high-frequency features through learnable filters, and selectively retains the most important high-frequency information.

4. We introduce a progressive training strategy to minimize GPU memory during training.

## 2 RELATED WORK

### 2.1 TRADITIONAL METHODS.

Due to the ill-posed nature of image deblurring, traditional methods Chen et al. (2020); Wen et al. (2021) primarily rely on hand-crafted priors to constrain the possible solutions. Recently, camera data from inertial measurement units has been leveraged to describe degradation parameters, pro-

viding guidance for blur kernel estimation Rong et al. (2024). While these priors can aid in blur removal, they often fail to accurately model the degradation process and lack generalizability.

## 2.2 CNN-BASED METHODS.

With the rapid progress of deep learning, many methods Pan et al. (2022); Ghasemabadi et al. (2024b) use deep CNNs to address image deblurring, eliminating the need for manually designed image priors. To better balance spatial details and contextual information, MPRNet Zamir et al. (2021) introduces cross-stage feature fusion to leverage features from multiple stages. IRNeXt Cui et al. (2023) rethinks convolutional network design, offering an efficient CNN-based architecture for image restoration. NAFNet Chen et al. (2022) evaluates baseline modules and suggests replacing nonlinear activation functions with multiplication, which simplifies the system's complexity. TURTLE Ghasemabadi et al. (2024a) employs a truncated causal history model for efficient, high-performance video restoration. CGNet Ghasemabadi et al. (2024b) integrates a global context extractor to effectively capture global information. FSNet Cui et al. (2024) uses multi-branch and content-aware modules to dynamically select the most relevant components. ELEDNet Kim et al. (2025) leverages cross-modal feature information with a low-pass filter to reduce noise while preserving structural details. MR-VNet Roheda et al. (2024) utilizes Volterra layers for efficient deblurring. While these methods are superior to traditional methods, the inherent limitations of convolutional operations hinder the models' ability to effectively capture long-range dependencies.

## 2.3 TRANSFORMER-BASED METHODS.

The transformer architecture Vaswani et al. (2017) has gained significant popularity in image deblurring Rao et al. (2025); Zhang et al. (2024) due to its content-dependent global receptive field, showing superior performance over traditional CNN-based baselines. However, image deblurring often deals with high-resolution images, and the attention mechanism in Transformers incurs quadratic time complexity, resulting in significant computational overhead. In order to reduce the computational cost, Uformer Wang et al. (2022), SwinIR Liang et al. (2021) and U$^2$former Feng et al. (2024) computes self-attention based on a window. Restormer Zamir et al. (2022), MRLPFNet Dong et al. (2023), and DeblurDiNAT Liu et al. (2024) compute self-attention across channels rather than in the spatial dimension, achieving linear complexity in relation to input size. However, the above methods inevitably cause feature loss. To this end, FFTformer Kong et al. (2023) explores the property of the frequency domain to estimate the scaled dot-product attention. For realistic image deblurring, HI-Diff Chen et al. (2023b) harnesses the power of diffusion models to generate informative priors, which are then integrated hierarchically into the deblurring process to improve results.

Although the methods mentioned above have achieved excellent performance through modular design, most overlook the fact that the degradation degrees varies across different regions of the blurred image. To enable differential processing of varying degrees of degradation, AdaRevD Mao et al. (2024) introduces a classifier to assess the degradation degree of image patches, but it relies on a limited number of predefined categories and a fixed patch size. This rigid approach reduces its effectiveness in adaptively managing degradation across patches of different sizes. In this paper, we propose an adaptively identifies blurred regions network for image deblurring, named AIBNet, which adept at differential handle regions with varying degrees of degradation

## 3 METHOD

In this section, we first provide an overview of the entire AIBNet pipeline. We then dive into the details of the proposed decoder. Lastly, we present the progressive training strategy.

### 3.1 OVERALL PIPELINE

Our proposed AIBNet, as shown in Figure 2 (a), includes a frozen encoder and $s$ sub-decoders. Each sub-decoder consists of $N$ SFDHBlocks and a HFSBlock. Given a degraded image $\mathbf{I} \in \mathbb{R}^{H \times W \times 3}$, AIBNet first uses a convolutional layer to extract shallow features $\mathbf{F} \in \mathbb{R}^{H \times W \times C}$, where $H$, $W$, and $C$ represent the height, width, and number of channels of the feature map, respectively. These shallow features are passed through a pre-trained encoder to produce encoder features $e^i$ (where $i =$

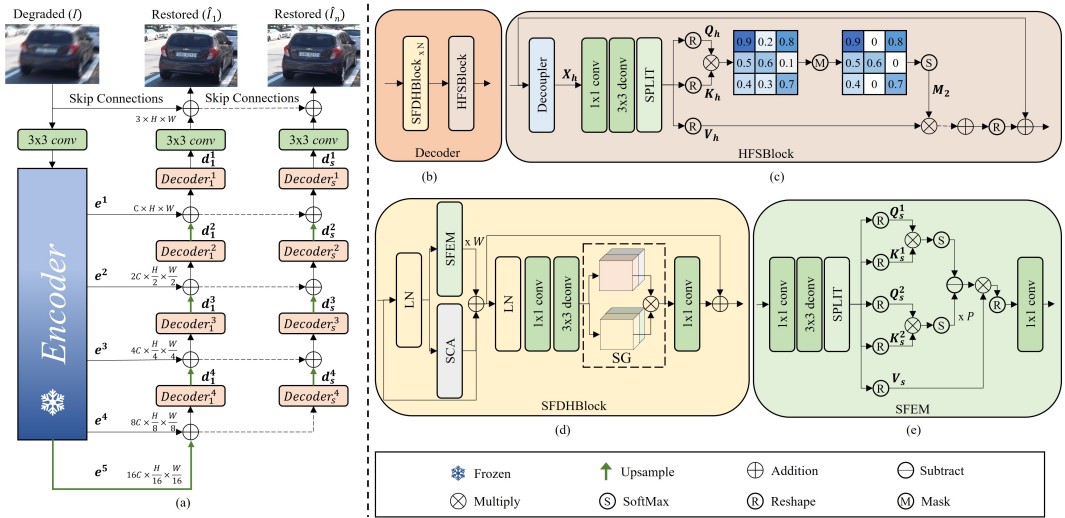

Figure 2: (a) The overall architecture of the proposed AIBNet. (b) The decoder, consisting of N spatial feature differential handling blocks (SFDHBlocks) and a high-frequency feature selection block (HFSBlock). (c) The structure of the HFSBlock, with the case of using a single mask matrix for simplicity. (d) The SFDHBlock, which consists of two branches: the SCA proposed in NAFNet Chen et al. (2022) and the spatial feature enhancement module (SFEM). (e) The structure of the SFEM.

$1, 2, 3, 4, 5$) at different scales. The encoder features are then fed into the decoder, which generates decoder features $d_s^i$ at different scales, progressively restoring them to their original size. It is important to note that, since our model includes multiple sub-decoders, the input to each subsequent sub-decoder is the output of the previous one. Finally, convolution is applied to the refined features to produce the residual image $\mathbf{X_s} \in \mathbb{R}^{H \times W \times 3}$ for $s_{th}$ sub-decoder. This residual image is added to the degraded image to produce the restored output: $\hat{\mathbf{I}}_\mathbf{s} = \mathbf{X_s} + \mathbf{I}$.

## 3.2 SPATIAL FEATURE DIFFERENTIAL HANDLING BLOCK

Most previous image deblurring methods achieve excellent performance by designing novel modules, but they often overlook the fact that the blur degree varies across different regions. To address this issue, we first design a spatial domain feature differential handling block (SFDHBlock), which helps the model focus on the key information of the blurred regions by removing the features of the non-blurred regions. As shown in Figure 2(d), given the input features at the $(l-1)_{th}$ block $X_{l-1}$, the procedures of SFDHBlock can be defined as:

$$
\begin{aligned}
X_l^{'} &= X_{l-1} + SCA(LN(X_{l-1})) + W \cdot SFEM(LN(X_{l-1})) \\
X_l &= X_l^{'} + f_{1x1}^c(SG(f_{3x3}^{dwc}(f_{1x1}^c(LN(X_l^{'})))))
\end{aligned}
\tag{1}
$$

where LN denotes Layer Normalization, $f_{3 \times 3}^{dwc}$ refers to the $3 \times 3$ depth-wise convolution, and $f_{1 \times 1}^c$ represents the $1 \times 1$ convolution. $W$ is the learnable parameter, which is directly optimized through backpropagation and initialized to 1. It's worth noting that our design is highly lightweight, as it does not introduce additional convolution layers. SCA stands for Simple Channel Attention, as proposed in NAFNet Chen et al. (2022). SFEM represents the spatial feature enhancement module, which is described below.

### 3.2.1 SPATIAL FEATURE ENHANCEMENT MODULE

Inspired by the theory of differential amplifier (**More proofs are given in the Appendix A**) which amplifies the difference of two input voltages with a fixed gain, and eliminates the interfering common mode signal by common mode rejection, we design the spatial feature enhancement module (SFEM). SFEM leverages feature differences to remove features from non-blurred regions and reduce implicit noise caused by intensive calculations, thereby helping the model focus on the key

information in the blurred regions. As shown in Figure 2(c), we first encode channel-wise context. Next, the feature is divided into five parts and reshaped to enable the subsequent attention calculation in the channel dimension, thereby reducing both time and memory complexity. Among these five features, we partition the query and key vectors into two groups and compute two separate $SoftMax$ attention maps. The result of subtracting these two maps is then used as the attention scores. Formally, given the features $X_{l-1}^n$ after LN, we can obtain the enhanced features $X_{l-1}^e$ by the following:

$$Q_s^1, K_s^1, Q_s^2, K_s^2, V_s = SPLIT(f_{3x3}^{dwc}(f_{1x1}^c(X_{l-1}^n)))$$

$$X_{l-1}^e = f_{1x1}^c(Reshape((SoftMax(\frac{Q_s^1 T(K_s^1)}{\beta}) - \alpha \cdot SoftMax(\frac{Q_s^2 T(K_s^2)}{\beta}))V_s)) \quad (2)$$

$\beta$ is a learning scaling parameter used to adjust the magnitude of the dot product before applying the $SoftMax$ function, and it is initialized as $\beta = \sqrt{C}$. $T$ denotes the transpose operation. $\alpha$ is the learnable scalar, initialized as:

$$\alpha = exp(\alpha_{Q_s^1} \cdot \alpha_{K_s^1}) - exp(\alpha_{Q_s^2} \cdot \alpha_{K_s^2}) + \alpha_{init} \quad (3)$$

where $\alpha_{Q_s^1}, \alpha_{K_s^1}, \alpha_{Q_s^2}, \alpha_{K_s^2}$ are the learnable parameters, which are directly optimized through back-propagation. And $\alpha_{init}$ a constant used for the initialization.

Finally, as shown in Eq. 1, the enhanced feature $X_{l-1}^e = SFEM(X_{l-1}^n)$ is fused with the other branches. This offers several advantages. First, SFEM addresses the limitation of SCA in modeling long-range dependencies. Second, SFEM enhances the features of blurred regions, making it easier for the model to focus on the most relevant information. Lastly, SFEM reduces the impact of implicit noise caused by intensive calculations through feature differences.

### 3.3 HIGH-FREQUENCY FEATURE SELECTION BLOCK

Based on the theory that the difference between blurred and sharp image pairs primarily lies in the high-frequency components Cui et al. (2023), we design the high-frequency feature selection block(HFSBlock) to further refine the identification of blurred regions in the frequency domain. The key motivation of our HFSBlock is to perform differential handling of different blurred regions. To achieve this, we do not design a new high-frequency feature capture module; instead, we use the existing Decoupler Cui et al. (2024) to dynamically generate high-frequency features $X_h$. These high-frequency features are then aggregated to leverage the sparsity by dynamically masking irrelevant features, thereby selecting the most important high-frequency components to retain for identifying blurred regions. For simplicity, we only show the case of using a single mask matrix in Figure 2(c). Specifically, given the output features $X_N$ of the $N_{th}$ SFDHBlock, we first dynamically generate the high-frequency features $X_h$. Similar to SFEM, we obtain the query $Q_h$, key $K_h$, and value $V_h$ matrices with the shape of $C \times H \times W$. Next, a dense attention matrix of shape $C \times C$ is generated by performing a dot-product operation between $Q_h$ and transposed $K_h$ across channels. Then, we selectively mask out the irrelevant elements to retain the most important high-frequency components for identifying blurred regions in the dense attention matrix.

In the example shown in Figure 2(c), we obtain the sparse attention matrix $M_2$ by keeping the first $\frac{2}{3}$ of the elements and setting the rest to 0 through masking. To maintain flexibility, we pass $n_m$ mask matrices and SoftMax to obtain $n_m$ sparse attention matrices $M_i(i = 1, 2, 3, ...n_m)$, and then perform a dot-product operation with $V_h$, respectively. Finally, the results are fused using learnable parameters and reshape to the original size to get the selection high-frequency features $X_{sh}$. The specific process is as follows:

$$M_i = SoftMax(Mask_i(\frac{Q_h T(K_h)}{\beta}))$$

$$X_{sh} = X_N + Reshape(\sum_{i=1}^{n_m} \lambda_i M_i \times V) \quad (4)$$

where $\lambda_i$ denotes the learnable parameters to control the dynamic selection of fusion. $Mask_i$ is the $i_{th}$ mask matrices, which defined as:

$$Mask_i(x) = \begin{cases} x, x \in fist\frac{i}{i+1}, \\ 0, otherwise. \end{cases} \quad (5)$$

As feature difference has already been applied in SFEM, the high-frequency feature values with large responses in HFSBlock no longer contain implicit noise. Therefore, HFSBlock retains the values of elements that align with the response and simply sets the elements that do not match the response to 0. The feature representation of blurred regions is enhanced both in the spatial domain by SFEM and in the frequency domain by HFSBlock, enabling our model to adaptively identify blurred regions for differential processing and accurate image deblurring.

### 3.4 PROGRESSIVE TRAINING STRATEGY

Since our model contains multiple sub-decoders, training it directly can be highly demanding on GPU memory. Additionally, to simplify the model and reduce computational complexity, we avoid introducing complex discriminative fusion mechanisms to connect the features of each sub-decoder. However, since the input of each subsequent sub-decoder depends heavily on the output of the previous one, the absence of such mechanisms can lead to issues like gradient collapse. To address this, we propose a progressive training strategy, where only one sub-decoder is trained at a time. After each sub-decoder is trained, its parameters are frozen before training the next one. This strategy offers multiple advantages. First, by training only one sub-decoder at a time, we significantly reduce the GPU memory requirements. Second, because each sub-decoder is trained with the actual image data, the input features for the next sub-decoder are more accurate, leading to better performance.

To optimize the proposed network AIBNet by minimizing the following loss function:

$$L = L_c(\hat{I}_s, \overline{I}) + \delta L_e(\hat{I}_s, \overline{I}) + \lambda L_f(\hat{I}_s, \overline{I}))$$
$$= \sqrt{||\hat{I}_s - \overline{I}||^2 + \epsilon^2} + \delta\sqrt{||\triangle\hat{I}_s - \triangle\overline{I}||^2 + \epsilon^2} + \lambda||\mathcal{F}(\hat{I}_s) - \mathcal{F}(\overline{I})||_1 \tag{6}$$

where $\overline{I}$ denotes the target image and $\hat{I}_s$ represents the output of the $s_{th}$ sub-decoder. $L_c$ refers to the Charbonnier loss with a constant of $\epsilon = 0.001$, while $L_e$ is the edge loss, where $\triangle$ denotes the Laplacian operator. $L_f$ represents the frequency domain loss, with $\mathcal{F}$ indicating the fast Fourier transform. To balance the contributions of the loss terms, we set the parameters $\lambda = 0.1$ and $\delta = 0.05$, as in Zamir et al. (2021); Cui et al. (2024).

## 4 EXPERIMENTS

In this section, we detail the experimental setup and provide both qualitative and quantitative comparisons. We also conduct ablation studies to demonstrate the effectiveness of our approach. (**More experiments are given in the Appendix A**)

### 4.1 EXPERIMENTAL SETTINGS

We use the Adam optimizer Kingma & Ba (2014) with parameters $\beta_1 = 0.9$ and $\beta_2 = 0.999$. The initial learning rate is set to $2 \times 10^{-4}$ and is gradually reduced to $1 \times 10^{-7}$ using the cosine annealing strategy Loshchilov & Hutter (2016). The networks are trained on $256 \times 256$ patches with a batch size of 32 for $4 \times 10^5$ iterations. Data augmentation includes both horizontal and vertical flips. For each decoder, we set $N$ (see Figure 2(b)) to 8. Additionally, we build 3 versions of AIBNet by varying the number of sub-decoders $s$ (see Figure 2(a)): AIBNet-S (1 sub-decoder), AIBNet-B (2 sub-decoders), and AIBNet-L (4 sub-decoders). For the encoder, we use UFPNet Fang et al. (2023).

### 4.2 EXPERIMENTAL RESULTS

#### 4.2.1 EVALUATIONS ON THE SYNTHETIC DATASET.

Table 1 presents the performance of various image deblurring methods on the synthetic GoPro Nah et al. (2016) and HIDE Shen et al. (2019) datasets. Overall, AIBNet outperforms competing methods, delivering higher-quality images with improved PSNR and SSIM values. Specifically, compared to the previous best method, AdaRevD-L Mao et al. (2024), our AIBNet-L achieves a 0.35 dB improvement on the GoPro dataset. Remarkably, even though our model was trained solely on the GoPro dataset, it still achieves state-of-the-art results (32.41 dB in PSNR) on the HIDE dataset, demonstrating its strong generalization ability. Performance further improves as the model size

Table 1: Quantitative evaluations of the proposed approach against state-of-the-art motion deblurring methods. Our AIBNet and AIBNet-B are trained only on the GoPro dataset Nah et al. (2016).

| Methods | GoPro | | HIDE | |
| --- | --- | --- | --- | --- |
| | PSNR ↑ | SSIM ↑ | PSNR ↑ | SSIM ↑ |
| UFPNet Fang et al. (2023) | 34.06 | 0.968 | 31.74 | 0.947 |
| MambaIR Guo et al. (2025) | 33.21 | 0.962 | 31.01 | 0.939 |
| ALGNet Gao et al. (2024) | 34.05 | 0.969 | 31.68 | 0.952 |
| MR-VNet Roheda et al. (2024) | 34.04 | 0.969 | 31.54 | 0.943 |
| FSNet Cui et al. (2024) | 33.29 | 0.963 | 31.05 | 0.941 |
| AdaRevD-L Mao et al. (2024) | 34.60 | 0.972 | 32.35 | **0.953** |
| XYScanNet Liu et al. (2025a) | 33.91 | 0.968 | 31.74 | 0.947 |
| PGDN Fang et al. (2025) | 34.17 | 0.950 | - | - |
| MDT Chen et al. (2025) | 34.26 | 0.969 | 31.84 | 0.948 |
| **AIBNet-S(Ours)** | 34.47 | 0.971 | 32.19 | 0.949 |
| **AIBNet-B(Ours)** | 34.69 | 0.972 | 32.35 | 0.952 |
| **AIBNet-L(Ours)** | **34.95** | **0.974** | **32.41** | **0.953** |

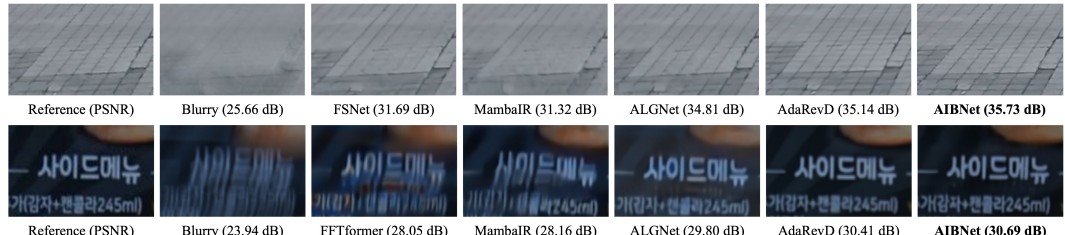

Figure 3: Image deblurring comparisons on the synthetic dataset Nah et al. (2016)(Top) and real-world dataset Rim et al. (2020)(Bottom).

increases (from AIBNet-S to AIBNet-L), emphasizing the scalability of our approach. Figure 3 showcases deblurred images from different methods, with our model's outputs being sharper and closer to the ground truth than those of other methods.

### 4.2.2 EVALUATIONS ON THE REAL-WORLD DATASET.

We further assess the performance of our AIBNet on real-world images from the RealBlur dataset Rim et al. (2020). Table 2 shows AIBNet achieves superior PSNR and SSIM scores. Specifically, compared to the previous best method, AdaRevD-L Mao et al. (2024), our approach improves PSNR by 0.25 dB on the RealBlur-R dataset and 0.13 dB on the RealBlur-J dataset. Figure 3 illustrates how our method effectively removes real blur while maintaining structural and textural details. In contrast, restored by other methods either appear overly smooth or fail to eliminate the blur.

Table 2: Quantitative evaluations on the real-word dataset RealBlur Rim et al. (2020).

| Methods | RealBlur-R | | RealBlur-J | |
| --- | --- | --- | --- | --- |
| | PSNR ↑ | SSIM ↑ | PSNR ↑ | SSIM ↑ |
| FFTformer Kong et al. (2023) | 40.11 | 0.973 | 32.62 | 0.932 |
| UFPNet Fang et al. (2023) | 40.61 | 0.974 | 33.35 | 0.934 |
| MambaIR Guo et al. (2025) | 39.92 | 0.972 | 32.44 | 0.928 |
| ALGNet Gao et al. (2024) | 41.16 | 0.981 | 32.94 | 0.946 |
| MR-VNet Roheda et al. (2024) | 40.23 | 0.977 | 32.71 | 0.941 |
| AdaRevD-L Mao et al. (2024) | 41.19 | 0.979 | 33.96 | 0.944 |
| **AIBNet-S(Ours)** | 41.12 | 0.980 | 33.88 | 0.956 |
| **AIBNet-B(Ours)** | 41.23 | 0.980 | 33.97 | 0.955 |
| **AIBNet-L(Ours)** | **41.44** | **0.981** | **34.09** | **0.958** |

Table 3: Ablation study on individual components of the proposed AIBNet.

| Net | Pre-trained | SFEM | HFSBlock | PSNR | △ PSNR |
|-----|-------------|------|----------|------|--------|
| (a) |             |      |          | 33.62 | - |
| (b) |             | ✔    |          | 33.93 | +0.31 |
| (c) |             |      | ✔        | 33.92 | +0.30 |
| (d) |             | ✔    | ✔        | 34.32 | +0.70 |
| (e) | ✔           | ✔    | ✔        | 34.47 | +0.85 |

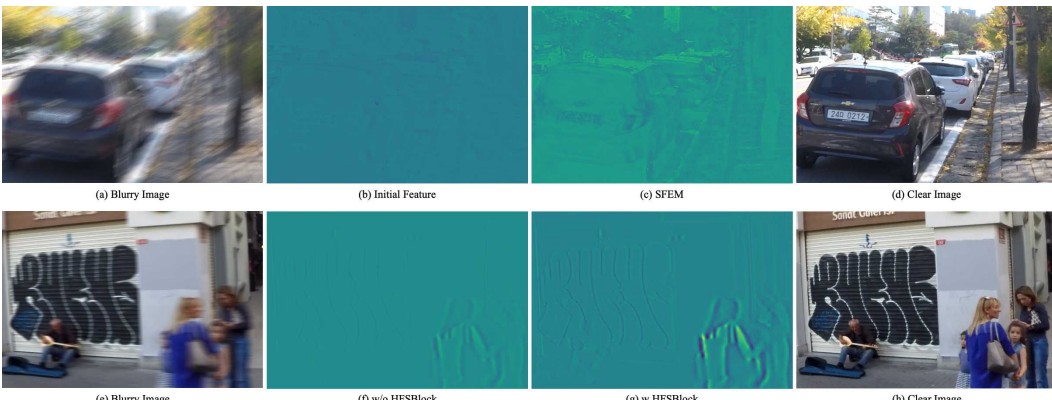

Figure 4: Internal features of SFEM and HFSBlock. Both modules capture finer details than the initial features. Please zoom in for a clearer view.

## 4.3 ABLATION STUDIES

### 4.3.1 EFFECTS OF INDIVIDUAL COMPONENTS

To assess the impact of each module, we use NAFNet Chen et al. (2022) as the baseline and progressively replace or add our proposed modules. As shown in Table 3(a), the baseline achieves a PSNR of 33.62 dB. Each module combination leads to a noticeable performance improvement. Specifically, adding the SFEM to NAFBlock improves performance by 0.31 dB (Table 3(b)). Incorporating the HFSBlock into the original NAFNet results in a significant increase in performance, boosting the PSNR from 33.62 dB to 34.92 dB (Table 3(c)). When both SFEM and HFSBlock are combined (Table 3(d)), our model achieves a 0.70 dB improvement over the original baseline. Finally, using a pre-trained model as the encoder and training only the decoder leads to a performance of 34.47 dB.

To further validate the effectiveness, we visualize the feature maps within it in Figure 4. In the initial features, main structures, such as the license plate number, are not well recovered before applying SFEM. In contrast, after the application of SFEM, the features are enhanced, allowing for a clearer representation of spatial details and more distincted structures. In HFSBlock, we selectively retain the most important high-frequency information. With our high-frequency feature selection, HFSBlock reveals finer details, such as pedestrians and graffiti on walls.

### 4.3.2 EFFECT OF THE NUMBER OF MASK MATRICES

The core component of HFSBlock is the mask matrix, which identifies the degraded regions by selectively preserving the most crucial high-frequency information and enhancing the frequency difference in those areas. To evaluate the impact of the number of mask matrices on model perfor-

Table 4: Effect of the number of mask matrices in HFSBlock.

| Number | 0 | 1 | 2 | 3 | 4 | 5 |
|--------|-----|-------|-------|-------|-------|-------|
| PSNR   | 34.22 | 34.35 | 34.40 | 34.43 | 34.47 | 34.46 |
| △ PSNR | -   | +0.13 | +0.18 | +0.21 | +0.25 | +0.24 |

Table 5: Effect of the pre-trained models.

| Net | Pre-trained | | Trainable | PSNR | △ PSNR |
|-----|-------------|--------|-----------|------|--------|
|     | NAFNet | UFPNet |        |      |        |
| (a) |        |        | ✔ | 34.74 | - |
| (b) | ✔      |        | ✔ | 34.89 | +0.15 |
| (c) | ✔      |        |   | 34.87 | +0.13 |
| (e) |        | ✔      | ✔ | 34.89 | +0.15 |
| (f) |        | ✔      |   | 34.95 | +0.17 |

Table 6: Effect of the progressive training strategy, where #$P$ denotes the parameters.

| Net | Fusion | Progressive | PSNR | △ PSNR | △ #$P$(M) |
|-----|--------|-------------|------|--------|-----------|
| (a) |        |        | 34.78 | - | - |
| (b) | SAM Zamir et al. (2021) |   | 34.86 | +0.08 | +6.12 |
| (c) | Fuse Mao et al. (2024)  |   | 34.84 | +0.06 | +1.32 |
| (d) |        | ✔      | 34.95 | +0.17 | -122.01 |

mance, we present the results in Table 4 for various configurations. As shown in Eq 5, the $i_{th}$ mask matrix retain the elements of the first $\frac{i}{i+1}$ and the others are set to 0. The performance is poorest at 34.22 dB when no mask matrix. However, performance improves as we incorporate different masking strategies with an increased number of mask matrices. The best performance is achieved with four mask matrices. Adding more mask matrices beyond this point results in a slight degradation in performance, as irrelevant or unnecessary feature representations are introduced.

### 4.3.3 EFFECTS OF THE PRE-TRAINED MODELS

Since our AIBNet utilizes an existing pre-trained model as the encoder, we evaluate the impact of different pre-trained models on performance. As shown in Table 5, the results vary with different pre-trained models (NAFNet, UFPNet), but all contribute to the model performance. Additionally, we examine how freezing the encoder parameters affects performance. From Table 5 (b) and (c), we observe that when NAFNet is used as the pre-trained model, freezing the encoder parameters leads to a decrease in performance. However, when UFPNet is used as the pre-trained model, freezing the parameters improves performance (see Table 5 (e) and (f)). Overall, freezing the encoder parameters has a minimal impact on performance. To save computational resources, we opt to freeze the encoder parameters and only train the decoder.

### 4.3.4 EFFECTS OF THE PROGRESSIVE TRAINING STRATEGY

As shown in Table 6(a), the worst performance occurs when multiple sub-decoders are trained directly. Adding a fusion module between the sub-decoders (Table 6(b) and (c)) improves performance, but also introduces additional parameters. When the progressive training strategy is applied (Table 6(d)), the performance is optimized. Moreover, since only one sub-decoder is trained at a time, our strategy is resource-efficient. Compared to direct training, the number of trainable parameters is drastically reduced by 122.01M.

## 5 CONCLUSION

In this paper, we propose an adaptively identifies blurred regions network (AIBNet) for image deblurring. Specifically, we design a spatial feature differential handling block (SFDHBlock) with the core being the spatial feature enhancement module (SFEM), which uses feature differences to help the model focus on key information. Additionally, we present a high-frequency feature selection block (HFSBlock), which uses learnable filters to extract and selectively retain the most important high-frequency features. To fully leverage the decoder's potential, we use a pre-trained model as the encoder and apply the above modules in the decoder. Finally, to reduce the resource burden during training, we employ a progressive training strategy. Extensive experiments show that AIBNet achieves superior performance.

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

# A  APPENDIX

## A.1  OVERVIEW

The Appendix is composed of:

Motivation Analysis A.2

Dataset A.3

More Ablation Studies A.4

Proofs of SFEM A.6

Additional Visual Results A.7

## A.2  MOTIVATION ANALYSIS

Although existing image deblurring methods Kong et al. (2023); Roheda et al. (2024) show strong performance, they overlook the varying degrees of blur across different regions. To tackle the first issue, AdaRevD Mao et al. (2024) introduces a classifier to assess the degradation level of image patches. However, AdaRevD Mao et al. (2024) relies on a fixed set of predefined categories and a constant blur patch size, limiting its adaptability to varying degradation levels across different patch sizes. This constraint also hinders its ability to address the second issue effectively. To achieve this, we propose AIBNet, a network that adaptively identifies blurred regions in both the spatial and frequency domains. Our AIBNet is primarily based on differential amplifier theory and the observation that the difference between blurred and sharp image pairs is concentrated in the high-frequency components. In the spatial domain, we use feature differences to remove non-blurred region features and reduce implicit noise caused by intensive calculations, allowing the model to focus on key information in the blurred regions. In the frequency domain, we selectively highlight the most important high-frequency information to emphasize the features of the degraded regions.

## A.3  DATASET

We evaluate the effectiveness of our method using the GoPro dataset Nah et al. (2016), which in-cludes 2,103 training image pairs and 1,111 evaluation pairs. To assess the generalizability of our model, we apply the GoPro-trained model to the HIDE Shen et al. (2019) dataset, consisting of

Table 7: The evaluation of model computational complexity between AdaRevD Mao et al. (2024) and our AIBNet.

| Net | AdaRevD-B Mao et al. (2024) | AdaRevD-L Mao et al. (2024) | AIBNet-S | AIBNet-B | AIBNet-L |
|---|---|---|---|---|---|
| MACs(G) | 348 | 461 | 114 | 228 | 456 |
| Trainable params(M) | 142.9 | 211.2 | 41.1 | 41.1 | 41.1 |
| PSNR(dB) | 34.50 | 34.60 | 34.47 | 34.69 | 34.95 |

Table 8: The replaceability of SFEM and HFSBlock.

| Net | PSNR | △ PSNR |
|---|---|---|
| Don't make changes | 34.47 | - |
| SFEM replaces HFSBlock | 34.31 | - 0.16 |
| HFSBlock replaces SFEM | 34.23 | - 0.24 |

2,025 images. Both the GoPro and HIDE datasets are synthetically generated. Additionally, we test our method on real-world images using the RealBlur Rim et al. (2020) dataset, which contains 3,758 training image pairs and 980 testing pairs, divided into two subsets: RealBlur-J and RealBlur-R.

## A.4 MORE ABLATION STUDIES

### A.4.1 AIBNET VS. ADAREVD MAO ET AL. (2024)

Both AIBNet and AdaRevD Mao et al. (2024) adopt the strategy of freezing the encoder and training multiple sub-decoders. To compare model complexity, we present a comparison in Table 7. As shown, thanks to our progressive training strategy, our model requires fewer parameters to train than AdaRevD Mao et al. (2024). Specifically, AIBNet-L uses only about $\frac{1}{5}$ of the parameters of AdaRevD-L Mao et al. (2024). Additionally, our MACs metric is lower, yet our performance is superior, highlighting the effectiveness of our model.

### A.4.2 THE REPLACEABILITY OF SFEM AND HFSBLOCK

In AIBNet, we use feature differencing in the spatial domain to remove the features of non-blurred regions, allowing the model to focus on key information in the degraded regions. In the frequency domain, the mask matrix is employed to selectively retain the most informative high-frequency features. Although both modules aim to select critical information, HFSBlock does not modify the parts with significant information in the spatial domain; it only sets the regions with small responses to zero, thus ensuring the accuracy of high-frequency features.

To demonstrate that our two modules cannot replace each other, we conducted experiments, as shown in Table 8. The results indicate that replacing one module with the other leads to performance degradation. The optimal performance is achieved only when SFEM is used in the spatial domain and HFSBlock is used in the frequency domain, as presented in this paper.

## A.5 SFEM VS. SCA CHEN ET AL. (2022)

To further validate the effectiveness of our designed SFEM in SFDHBlock, we compare the feature map visualizations between SCA Chen et al. (2022) and SFEM in Figure 5. As shown, the feature map information generated by our SFEM is more accurate than that of the SCA branch, indicating that our module effectively helps the model focus on the key features of the blurred areas, thereby improving restoration performance.

### A.5.1 ALTERNATIVE OF HFSBLOCK

To further validate the design advantage of our HFSBlock, we replace it with an existing sparse attention method Chen et al. (2023a); Xiao et al. (2024). The experimental results, shown in Table 9, demonstrate that the performance is optimal when our HFSBlock is used.

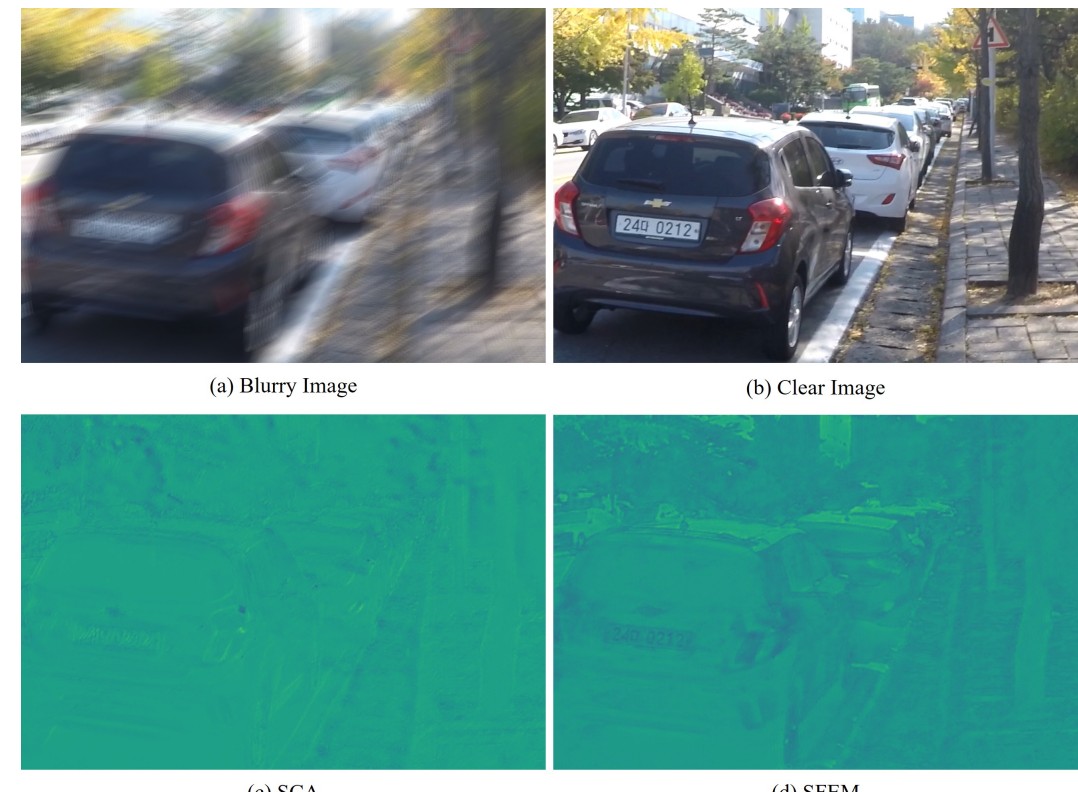

(a) Blurry Image          (b) Clear Image

(c) SCA          (d) SFEM

Figure 5: Comparison of feature maps between SCA and SFEM.

Table 9: The replaceability of SFEM and HFSBlock.

| Net | PSNR | △ PSNR |
|---|---|---|
| HFSBlock | 34.47 | - |
| TKSA Chen et al. (2023a) | 34.35 | - 0.12 |
| TTSA Xiao et al. (2024) | 34.36 | - 0.11 |

### A.5.2 RESOURCE EFFICIENT

We evaluate the model complexity of our proposed approach and other state-of-the-art methods in terms of running time and MACs. As shown in Table 10, our method achieves the lowest MACs value while delivering competitive performance in terms of running time. However, due to the inclusion of multiple sub-decoders, the complexity of our system is relatively high, reaching 114G MACs. Nonetheless, by leveraging the progressive training strategy introduced in this paper, the training process remains resource-efficient, requiring less computational power.

Table 10: The evaluation of model computational complexity.

| Method | Time(s) | MACs(G) | PSNR↑ | SSIM↑ |
|---|---|---|---|---|
| MPRNet | 1.148 | 777 | 32.66 | 0.959 |
| MambaIR Guo et al. (2025) | 0.743 | 439 | 33.21 | 0.962 |
| AdaRevD-L Mao et al. (2024) | 0.761 | 460 | 34.60 | 0.972 |
| **AIBNet-S(Ours)** | **0.241** | **114** | 34.47 | 0.971 |
| **AIBNet-B(Ours)** | 0.552 | 228 | 34.69 | 0.972 |
| **AIBNet-L(Ours)** | 0.729 | 456 | **34.95** | **0.974** |

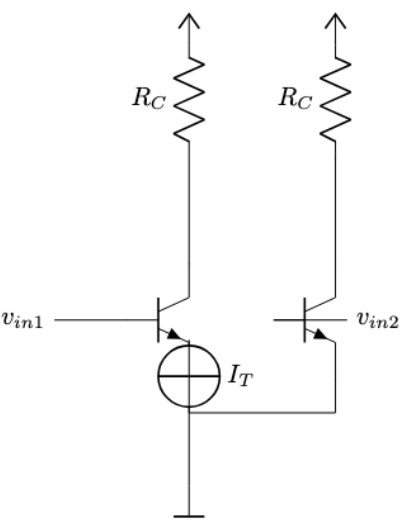

Figure 6: Differential amplifier circuit.

### A.6 PROOFS OF SFEM

Inspired by the theory of differential amplifiers, where the difference between two signals cancels out common-mode noise, we design the spatial feature enhancement module (SFEM). SFEM leverages feature differences to remove features from non-blurred regions and reduce implicit noise caused by intensive calculations, thereby helping the model focus on the key information in the blurred regions.

As shown in Figure 6 differential amplifier amplifies the difference between two input voltages while rejecting any voltage common to both inputs.

**Assumptions**

- The transistors are perfectly matched.
- The current source $I_T$ is ideal and constant.
- Small-signal analysis is used (i.e., linear region of operation).
- $v_{in1}$ and $v_{in2}$ are differential inputs.

**Differential Input and Output**

Define the input voltages as:

$$v_{in1} = V_{CM} + \frac{v_d}{2}, \quad v_{in2} = V_{CM} - \frac{v_d}{2}$$

where $v_d$ is the differential input voltage and $V_{CM}$ is the common-mode voltage.

Due to symmetry, under differential excitation, the total tail current $I_T$ is evenly split:

$$I_{C1} = I_{C2} = \frac{I_T}{2}$$

when $v_d = 0$. Under small differential input $v_d$, and assuming the transconductance $g_m$ of each transistor is:

$$g_m = \frac{I_C}{V_T}$$

where $V_T$ is the thermal voltage, and $I_C = I_T/2$, the small-signal collector currents are:

$$\Delta i_{C1} = g_m \cdot \left(\frac{v_d}{2}\right), \quad \Delta i_{C2} = -g_m \cdot \left(\frac{v_d}{2}\right)$$

**Output Voltage Derivation**

The output voltage at each collector is:

$$v_{o1} = -R_C \cdot \Delta i_{C1} = -R_C \cdot g_m \cdot \frac{v_d}{2}$$

$$v_{o2} = -R_C \cdot \Delta i_{C2} = R_C \cdot g_m \cdot \frac{v_d}{2}$$

Differential output voltage:

$$v_{out} = v_{o2} - v_{o1} = R_C \cdot g_m \cdot v_d$$

The SFEM is processed according to the differential amplifier principle, we first encode channel-wise context by applying 1×1 convolutions followed by 3×3 depth-wise convolutions. Next, the feature is divided into five parts and reshaped to enable the subsequent attention calculation in the channel dimension, thereby reducing both time and memory complexity. Among these five features, we partition the query and key vectors into two groups and compute two separate $SoftMax$ attention maps. The result of subtracting these two maps is then used as the attention scores. Formally, given the features $X_{l-1}^n$ after LN, we can obtain the enhanced features $X_{l-1}^e$ by the following:

$$
\begin{aligned}
X_{l-1}^c &= f_{3x3}^{dwc}(f_{1x1}^c(X_{l-1}^n)) \\
Q_s^1, K_s^1, Q_s^2, K_s^2, V_s &= SPLIT(X_{l-1}^c) \\
att1 &= SoftMax(\frac{Q_s^1 T(K_s^1)}{\beta}) \\
att2 &= SoftMax(\frac{Q_s^2 T(K_s^2)}{\beta}) \\
X_{l-1}^e &= f_{1x1}^c(Reshape((att1 - \alpha \cdot att2)V_s))
\end{aligned}
\tag{7}
$$

$\beta$ is a learning scaling parameter used to adjust the magnitude of the dot product before applying the $SoftMax$ function, and it is initialized as $\beta = \sqrt{C}$. $T$ denotes the transpose operation. $\alpha$ is the learnable scalar, initialized as:

$$\alpha = exp(\alpha_{Q_s^1} \cdot \alpha_{K_s^1}) - exp(\alpha_{Q_s^2} \cdot \alpha_{K_s^2}) + \alpha_{init} \tag{8}$$

where $\alpha_{Q_s^1}, \alpha_{K_s^1}, \alpha_{Q_s^2}, \alpha_{K_s^2}$ are the learnable parameters, which are directly optimized through back-propagation. And $\alpha_{init}$ a constant used for the initialization.

## A.7 ADDITIONAL VISUAL RESULTS

In this section, we present additional visual results alongside state-of-the-art methods to highlight the effectiveness of our proposed approach, as shown in Figures 7 and 8. It is evident that our model produces more visually appealing outputs for both synthetic and real-world motion deblurring compared to other methods.

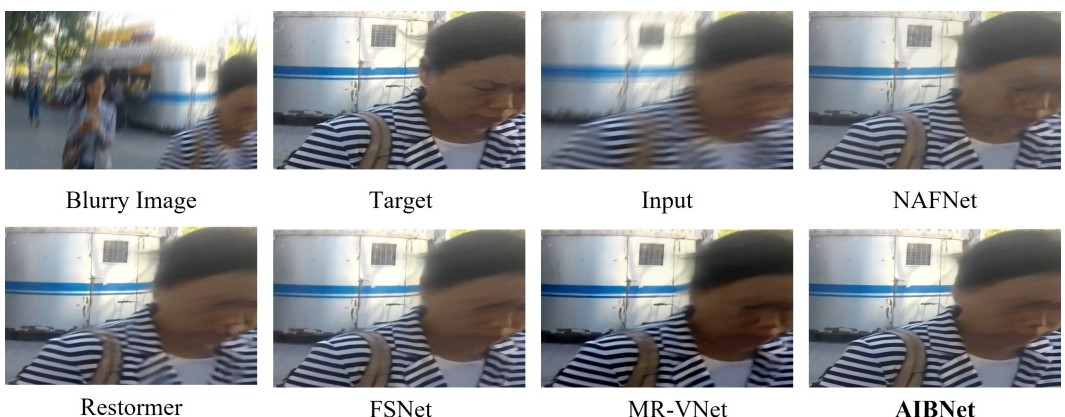

| Blurry Image | Target | Input | NAFNet |
| :---: | :---: | :---: | :---: |

| Restormer | FSNet | MR-VNet | **AIBNet** |
| :---: | :---: | :---: | :---: |

Figure 7: Comparison of image motion deblurring on the HIDE dataset Shen et al. (2019).

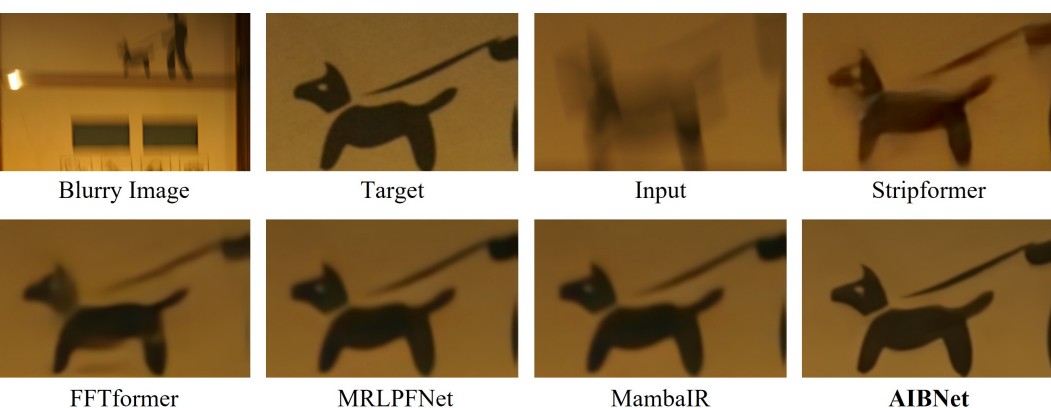

| Blurry Image | Target | Input | Stripformer |
| :---: | :---: | :---: | :---: |

| FFTformer | MRLPFNet | MambaIR | **AIBNet** |
| :---: | :---: | :---: | :---: |

Figure 8: Comparison of image motion deblurring on the RealBlur dataset Rim et al. (2020).

