# OpenReview forum: "Adaptive Identification of Blurred Regions for Accurate Image Deblurring"
_ICLR.cc/2026/Conference — ICLR 2026 Conference Withdrawn Submission_

### Official Review · Reviewer_pVBJ · 2025-10-24

**Soundness:** 2
**Presentation:** 3
**Contribution:** 2
**Rating:** 4
**Confidence:** 5

**Summary:**

This paper proposes AIBNet, a network that adaptively identifies the blurred regions, enabling differential restoration of these regions. The authors design a spatial feature differential handling block (SFDHBlock), with the core being the spatial domain feature enhancement module (SFEM) to focus on the key information in the blurred regions and eliminate the interference of implicit noise. Additionally, they propose a high frequency feature selection block (HFSBlock) that first uses learnable filters to extract high-frequency features and then selectively retains the
most important ones. Extensive experiments demonstrate that our AIBNet achieves superior performance in image deblurring.

**Strengths:**

1. The motivation and the proposed modules, including SFDHBlock and HFSBlock, are well-reasoned and appropriate for the image deblurring task.

2. The proposed architecture is elegant and easy to follow.

3. The paper is well written and clearly presented.

**Weaknesses:**

1. This work appears to be primarily an engineering effort that lacks sufficient novelty in the field of image deblurring. The proposed modules, including the use of a pre-trained encoder, SFDHBlock, and HFSBlock, are relatively straightforward ideas aimed at improving performance.

2. The proposed SFEM, which subtracts two attention maps, is quite similar to the "Differential Transformer" published at ICLR 2025, which reduces the novelty and contribution of SFEM.

3. Since the overall architecture seems quite large, the authors should report the total number of parameters, FLOPs, and inference time of the proposed AIBNet, including AIBNet-S, AIBNet-B, and AIBNet-L, and to compare these metrics against competing methods.

**Questions:**

1. The author should provide the total number of parameters, FLOPs, and inference time of the proposed AIBNet, including AIBNet-S, AIBNet-B, and AIBNet-L, and to compare these metrics against competing methods.

2. I am curious about the main differences between the Differential Transformer published at ICLR 2025 and the proposed SFEM. The underlying ideas appear to be almost identical.

---

### Official Review · Reviewer_y3eP · 2025-10-26

**Soundness:** 2
**Presentation:** 2
**Contribution:** 2
**Rating:** 4
**Confidence:** 5

**Summary:**

This paper aims to address the problem that existing image deblurring methods treat all regions with a uniform degradation level, despite the fact that different areas of an image often experience varying degrees of blur. The authors propose the AIBNet framework, which adaptively identifies blurred regions in both the spatial and frequency domains and performs differential restoration accordingly. The core modules include the Spatial Feature Differential Handling Block (SFDHBlock) — where the Spatial Feature Enhancement Module (SFEM) enhances blurred-region representations via feature differencing; the High-Frequency Feature Selection Block (HFSBlock) — which employs learnable mask matrices to select key high-frequency components; and a Progressive Training Strategy designed to reduce memory consumption in multi-decoder architectures.

**Strengths:**

AIBNet introduces a novel architecture that combines spatial-domain differencing and frequency-domain feature selection to distinguish blurred regions. The SFEM integrates the principle of a differential amplifier into visual feature modeling, offering theoretical insight. The proposed model achieves superior performance across multiple datasets (e.g., GoPro and HIDE), while the progressive training strategy reduces computational and memory overhead.

**Weaknesses:**

However, a major weakness of this work lies in the discrepancy between its stated goal and experimental evidence. The title and abstract emphasize “Adaptive Identification of Blurred Regions”, yet the authors provide no direct proof that their method can indeed distinguish blurred regions. For example, there are no visualizations or heatmaps illustrating which parts of an image are identified as blurred versus sharp. This omission significantly undermines the paper’s core claim and overall persuasiveness.

Sections 3.2 and 3.3 describe the SFEM and HFSBlock separately, but the paper fails to clarify their sequential or parallel relationship, nor does it discuss how the output of SFEM influences or interacts with the input of HFSBlock.

In Table 6, only the change in parameter count (Δ#P) is reported, without inference time or FLOPs. Similarly, Tables 1–2 omit GPU memory consumption and runtime latency, preventing a fair assessment of computational efficiency.

The experimental section lacks an ablation study for the progressive training strategy (Sec. 3.4); there is no comparison with a baseline trained end-to-end in a single stage. Moreover, the effect of the SCA branch (inherited from NAFNet) has not been examined, nor has the performance scaling with different numbers of sub-decoders s (1, 2, 4) been systematically analyzed.

Finally, in Eqs. (2)–(3) (Sec. 3.2.1), the SoftMax differencing formulation and the parameters α and β lack theoretical justification or stability analysis. In Eqs. (4)–(5), the definition of the masking function (“first i/(i+1)”) is vague, and no explanation is given for the principle behind its sparsification behavior. These issues collectively fall short of the mathematical rigor expected at ICLR standards.

**Questions:**

See Weakness

---

### Official Review · Reviewer_jAQS · 2025-10-30

**Soundness:** 3
**Presentation:** 3
**Contribution:** 2
**Rating:** 4
**Confidence:** 4

**Summary:**

This paper proposed to solve the spatial variant motion blurs with high frequency enhancement module and non-blurred region filtering module.

**Strengths:**

1. The paper shows good results on the GoPro, HIDE and RealBlur datasets.
2. Progressive decoder reduced the training burden

**Weaknesses:**

1. Please write T to the top right corer of Q. It is misleading in Eqn. 2 and 4.
2. Performance heavily relies on a pre-trained encoder.
3. The mathematical analysis of SFEM is not enough.

**Questions:**

1. Authors claim that "To fully leverage the potential of the decoder, we use a pre-trained model as the encoder and adopt multiple sub-decoders." I do not get the causal relationship between fully leveraging docoder and using pretrained encoders.
2. I think AdaRevD is not the only one that handled spatially varying degradations. There are still quite a few. Though some are cited for example Rong et al. (2024). Fang et al. (2025), no  description about the difference of these methods and how is the proposed method better than them.
3. There is no explicit supervision in SFEM. How can you make sure that Eqn. 2 correctly enhanced blurred regions and filtered non-blurred regions. Any formal mathematical justification in computer vision perspective rather than circuit?
4. How does the method handle the spatial variant blur. To me, I can only see that the model separate blurred and non-blurred region. But it is unclear for regions with different blurs.

---

### Official Review · Reviewer_Q1qr · 2025-10-31

**Soundness:** 3
**Presentation:** 2
**Contribution:** 2
**Rating:** 2
**Confidence:** 5

**Summary:**

This paper presents a network for image motion deblurring that adaptively identifies blurred regions. To achieve this, the authors propose two key modules, the SFEM and HFSBlock. In addition, a progressive training strategy is introduced to further enhance performance. The proposed model is evaluated on synthetic and real-world datasets and achieves promising performance.

**Strengths:**

1. A spatial feature differential handling block is introduced to enable the model to focus on key information in the blurred regions.

2. A high-frequency feature section block is proposed to retain the most important high-frequency regions.

3. Technically, a progressive training strategy is used to save GPU memory and leads to performance improvements.

4. The model achieves promising performance on both synthetic and real-world datasets.

**Weaknesses:**

1. The claim `most overlook the fact that the degradation degree varies across different regions` appears inappropriate. This topic has been extensively studied in recent years and is commonly referred to as spatially variant degradation [1].

2. Several recent and important references are missing from the comparative analysis, such as the Mamba-based deblurring method EVSSM [2]. In addition, compared with EVSSM, the proposed model in this paper involves a larger number of parameters.

3. The novelty of the proposed method is limited, as it primarily combines several existing techniques. In particular, the SFEM module appears to be derived from the **Differential Transformer** [3]; however, this prior work is not cited in the manuscript.

Refs.

[1] Dynamic Scene Deblurring Using Spatially Variant Recurrent Neural Networks, CVPR18.

[2] Efficient visual state space model for image deblurring, CVPR25.

[3] Differential Transformer, ICLR25.

**Questions:**

Could the authors clarify the meanings of `fist` in Eq. (5) and `SG` in Eq. (1)?

---

### Note · Authors · 2025-11-12

I have read and agree with the venue's withdrawal policy on behalf of myself and my co-authors.